# Pivotal Role of Ubiquitin Carboxyl-Terminal Hydrolase L1 (UCHL1) in Uterine Leiomyoma

**DOI:** 10.3390/biom13020193

**Published:** 2023-01-18

**Authors:** Tomoo Suzuki, Yidan Dai, Masanori Ono, Junya Kojima, Toru Sasaki, Hiroshi Fujiwara, Naoaki Kuji, Hirotaka Nishi

**Affiliations:** 1Department of Obstetrics and Gynecology, Tokyo Medical University, Tokyo 160-0023, Japan; 2Department of Obstetrics and Gynecology, Graduate School of Medical Sciences, Kanazawa University, Kanazawa 920-8641, Japan

**Keywords:** collagen production, LDN57444, leiomyoma, ubiquitin carboxyl-terminal hydrolase L1, UCHL1

## Abstract

Uterine leiomyomas are smooth-muscle tumors originating in the myometrium and are the most common pelvic tumors in women of reproductive age. Symptomatic tumors may result in abnormal uterine bleeding, bladder dysfunction, pelvic discomfort, and reproductive issues, such as infertility and miscarriage. There are currently few non-invasive treatments for leiomyoma, but there are no practical early intervention or preventive methods. In this study, human uterine leiomyoma and myometrial tissues were used to detect the protein and mRNA expression levels of UCHL1. To explore the effects of UCHL1 knockdown and inhibition in leiomyoma and myometrial cells, we determined the mRNA expressions of COL1A1 and COL3A1. Collagen gel contraction and wound-healing assays were performed on myometrial and leiomyoma cells. We found that UCHL1 expression was considerably higher in uterine leiomyomas than in the myometrium. COL1A1 and COL3A1 expression levels were downregulated after inhibition of UCHL1 in human leiomyoma cells. Furthermore, the elimination of UCHL1 significantly decreased the migration and contractility of leiomyoma cells. In conclusion, these results indicate that UCHL1 is involved in the growth of leiomyoma in humans. For the treatment of uterine leiomyoma, targeting UCHL1 activity may be a unique and possible therapeutic strategy.

## 1. Introduction

Despite that studies on the epidemiology of leiomyoma have been published, estimates of their incidence and prevalence greatly vary depending on the method of diagnosis and the population under study [1,2,3]. Additionally, a wide range of risk factors, including biological, demographic, reproductive, and lifestyle factors, have been linked to the development of leiomyoma [1,2,4]. Therefore, the actual incidence and prevalence of leiomyoma remains unknown. It usually grows throughout the reproductive age, has a clonal origin, and shrinks after menopause. In addition to related reproductive diseases such as infertility and miscarriage, uterine leiomyomas can cause abnormal uterine bleeding, bladder dysfunction, and pelvic discomfort [5]. To date, uterine leiomyomas constitute the most common cause for hysterectomy in women of reproductive age [6,7]. Surgery is still the main therapeutic option, since the processes that cause uterine leiomyoma growth remain poorly understood.

Cellular homeostasis depends on the ubiquitin-proteasome system. Three enzymes, E1, E2, and E3, control a reversible protein modification known as ubiquitination, which is the process of adding ubiquitin molecules to lysine residues in a protein. Most proteins target the 26S proteasome for degradation after ubiquitination [8]. Deubiquitinases are a broad class of enzymes that eliminate ubiquitin modifications. More than 90 deubiquitinases have been identified in humans. There are four members of the ubiquitin C-terminal hydrolase subfamily of deubiquitinases: BRCA1-associated protein-1, UCHL1, UCHL3, and UCHL5 [9,10]. According to previous reports, UCHL1 is frequently overexpressed in neurons and highly expressed in certain cancer cells [9]. In various heart diseases, UCHL1 expression is upregulated in abnormal fibroblasts, which may be a significant response to pathological heart remodeling and dysfunction [8,10]. In abnormal cardiac fibrosis, UCHL1 is a key regulator that induces collagen I and CTGF expression [11]. However, the function of UCHL1 in the uterus remains unclear.

UCHL1 is reportedly expressed in oocytes and is essential for the development of the ovary and female fertility [12]. Increased expression of UCHL1 has been shown in morphologically abnormal oocytes in prepubertal mouse ovaries [12,13]. However, decreased UCHL1 expression affects ovarian function, hormonal stimulation response, and estrous cyclicity. These results suggest that an appropriate level of UCHL1 expression in the ovary is critical for fertility. It is also well-known that UCHL1 has dual specificity for ubiquitin and neural precursor cell expressed, developmentally downregulated 8 (NEDD8), a protein encoded by the NEDD8 gene which shares 60% amino acid similarity with ubiquitin [14]. NEDD8 is removed from protein conjugates using a variety of proteases, including UCHL1. NEDD8 conjugates to its target proteins by a process known as neddylation (referred to as NEDDylation). Neddylation, a post-translational modification that involves the addition of the ubiquitin-like protein NEDD8 to substrate proteins, affects a variety of biological processes, including the advancement of the cell cycle, control of the cytoskeleton, and the development of tumors. Aberrant neddylation is known to cause leiomyoma and liver fibrosis [15,16].

Here, we found that UCHL1 is aberrantly upregulated in leiomyomas. Leiomyoma cell migration, gel contraction, and collagen synthesis considerably improved after the administration of the UCHL1 inhibitor LDN57444. In this study, we investigated the mechanisms by which UCHL1 contributed to the pathology of uterine leiomyoma and showed a unique process of leiomyoma development by UCHL1.

## 2. Materials and Methods

### 2.1. Patients

In this study, we used tissue samples obtained from patients hospitalized in the Department of Obstetrics and Gynecology at the Tokyo Medical University Hospital. The study participants involved women qualified for surgical treatment for hysterectomy of multiple uterine leiomyomas without adenomyosis or malignancies. Seventeen premenopausal women provided tissues from their myometrium and leiomyomas. We performed DNA sequencing and investigated the mutation status of MED12 in the 17 leiomyoma specimens used in this experiment. As a result, in this study, it was found that 6 out of 17 (35.29%) had MED12 mutations (Appendix A). We also tested the high mobility group AT-hook (HMGA2) overexpression in our specimens. Immunohistochemistry was used to evaluate HMGA2 overexpression [17]. Only samples that showed strong positivity in the immunoreaction were considered positive. It was found that 3 out of the 17 (17.6%) specimens showed HMGA2 mutations (Appendix A).

### 2.2. Study Approval

The study was conducted in accordance with the Declaration of Helsinki and approved by the Institutional Review Board of Tokyo Medical University (protocol code TS2021-0188) for studies involving humans.

### 2.3. Immunohistochemistry

After antigen retrieval, 5 µm-thick slices of formalin-fixed, paraffin-embedded tissue were subjected to immunohistochemical staining using an anti-UCHL1 mouse monoclonal antibody (1:33, human recombinant UCH-L1, R&D System. Inc., Minneapolis, MN, USA). DAB was used to see the signal (Fujifilm Wako Pure Chemical Corporation, Osaka, Japan). Hematoxylin was used as the counterstain. Images were obtained using a whole-slide scanner (NanoZoomer XR, Hamamatsu Photonics, Hamamatsu, Japan).

### 2.4. RNA Isolation and Quantitative Real-Time PCR Analysis

Total RNA was extracted from tissues and cell cultures using the AllPrep DNA/RNA/Protein mini kit (Qiagen, Germantown, MD, USA) according to the manufacturer’s protocol. RNA concentration was determined using a NanoDrop 2000 Spectrophotometer (Thermo Fisher Scientific, Waltham, MA, USA). A total of 30 ng (per tissue) or 700 ng (per cell line) of RNA was subjected to reverse transcription for the synthesis of single-stranded cDNA using the Superscript IV VILO kit following the manufacturer’s protocol (Applied Biosystems, Foster City, CA, USA). We used the power SYBR Green PCR master mix (Applied Biosystems) for nucleic acid quantification in real-time PCR. The reactions were incubated for 10 min at 95 °C, followed by 40 cycles of 15 s at 95 °C and 1 min at 60 °C. The levels of mRNA expression were determined using the StepOne System (Thermo Fisher Scientific, Waltham, MA, USA), with ACTB as an internal control. All reactions were run in duplicate. Relative expression was analyzed using the comparative cycle threshold method (2^−ΔΔCt^). Values are expressed as fold-changes compared to the control group. The primer pairs used for quantitative real-time PCR are listed in Table 1.

### 2.5. Protein Isolation and Western Blotting Analysis

Proteins were extracted using an AllPrep DNA/RNA/Protein Mini Kit (Qiagen, Germantown, MD, USA) according to the manufacturer’s protocol. Protein concentrations were quantified using a Pierce BCA Protein Assay Kit (Thermo Scientific) at 562 nm using a PerkinElmer EnSpire plate reader. Proteins (20 µg) were mixed with NuPAGE 4× lithium dodecyl sulfate sample buffer (Invitrogen) and NuPAGE 10× reducing agent and then heated at 70 °C for 10 min before centrifugation at 17,500 rpm for 10 min at 4 °C. Proteins were separated on a 4–12% Bis-Tris-polyacrylamide gel (Invitrogen) and transferred to an iBlot Mini Transfer Stack PVDF membrane (Invitrogen). The membranes were probed with antibodies against UCHL1 (1:2000; MAB6007; R&D Systems, Minneapolis, MN, USA) or β-actin (1:5000; MAB1501R; Millipore, Bedford, MA, USA) overnight at 4 °C after blocking with 5% skim milk in Tris-buffered saline Tween 20 for 1 h. The secondary antibody anti-mouse IgG, HRP-linked (1:1000 or 1:2000; NA931-1ML, Cytiva, Westborough, MA, USA) was used. Immune complexes were visualized using chemiluminescence (EMD Millipore, cat. no. WBLUF0100, cat. no. WBLUC0100) and quantified using a ChemiDoc XRS Plus system with Image Lab software (Bio-Rad, Hercules, CA, USA). The primary antibodies used in this study are listed in Table 2.

### 2.6. Cell Culture

Small portions of leiomyomas and matched myometrium of 12 patients were used to isolate leiomyomas and myometrial cells. Tissues were obtained during surgery from women (range 37–55 years) undergoing hysterectomy, in addition to some basic endocrine information (e.g., day of menstrual cycle, parity). Written informed consent was obtained from each patient and the use of human tissue specimens was approved by the Institutional Review Board for Human Research at Tokyo Medical University (T2021-0267). None of these cases had any previous history of uterine cancer, and all samples were confirmed by histopathological examination to be free of malignancy. The myometrial tissue was manually cut into small pieces of <1 mm^3^, which were then incubated for 4 h in DMEM/F12 medium (Gibco, Grand Island, NY, USA) containing 0.2% (wt./vol) collagenase (Wako), 0.05% DNase I (Invitrogen), and 1% antibiotic-antimycotic mixture (Invitrogen) at 37 °C on a shaker. After shaking, the digested tissue was filtered through a sterile gauze (Hakuzo Medical, Japan) to remove undigested tissues. The isolated leiomyoma and myometrial cells were cultured in DMEM/F12 medium (Gibco) supplemented with 10% FBS and 1% glutamine in 5% CO_2_ at 37 °C. We confirmed that freshly isolated first-passage cultures of leiomyoma cells display smooth-muscle markers by conducting immunofluorescence using anti-desmin, vimentin, and α-smooth-muscle actin (ACTA2) antibodies (Appendix A).

### 2.7. UCHL1 shRNA Transduction and Cell Treatment

UCHL1 knockdown was performed by transducing human leiomyoma and myometrial cells with control or UCHL1 short-hairpin RNA (shRNA) lentiviral virus particles (sc-108080 and sc-42304-V, Santa Cruz Biotechnology, Dallas, TX, USA) using TransDux MAX (LV860A-1, System Biosciences, Palo Alto, CA, USA), according to the manufacturer’s protocol. Briefly, leiomyoma and myometrial cells (50,000 cells per well) were seeded in a 24-well plate in DMEM/F12 with 10% FBS the day before transduction until 50–70% confluent. After aspirating the medium from the cells, the transduction solution and lentiviral virus particles were added to each well and incubated for 72 h at 37 °C in 5% CO_2_. Leiomyoma and myometrial cells with control shRNA or UCHL1 shRNA were subjected to RNA extraction, followed by quantitative real-time PCR analysis.

### 2.8. UCHL1 Inhibition and Cell Treatment

The small-molecule UCHL1 inhibitor LDN57444 was used to assess whether the inhibition of UCHL1 could prevent leiomyoma growth. Leiomyoma and myometrial cells were treated with DMSO (control) or 10 µmol/mL of LDN57444 (Tocris Bioscience, Bristol, UK) for five days.

### 2.9. Collagen Gel Contraction Assay

Leiomyoma and myometrial cells transduced with control or shUCHL1 were suspended in 500 µL of Cellmatrix type IA collagen solution (1.8 × 10^5^ cells/mL, Nitta Gelatin, Osaka, Japan) in a culture medium in a 24-well plate and allowed to gelate for 30 min at 37 °C. After gelation, 1 mL of DMEM/F12, supplemented with 10% FBS, was added to each well. Cells developed in a three-dimensional collagen gel matrix, which offers a favorable environment for cellular growth. The cell–extra cellular matrix combination began to compress when cells engaged with extracellular matrix collagen, and the gel gradually shrunk. The diameter of the gel decreased as the contraction increased. After incubation for 96 h, the collagen gels were photographed, and the surface area of the gel was measured.

### 2.10. Wound-Healing Assay

Leiomyoma and myometrial cells, transduced with control or shUCHL1 for 72 h, were maintained in DMEM/F12 with 10% FBS in 6-well plates at a density of 4 × 10^5^ cells per well for 24 h at 37 °C in 5% CO_2_. A wound was made by gently scratching the monolayer cell in a straight line with a sterile P-200 pipette tip in the center of each well, and the floating cells and debris were removed by washing with sterile 1× PBS twice. Serum-free medium was added to the wells, and the cells were grown overnight (at 37 °C in 5% CO_2_). The cells were imaged in each well using a microscope (EVOS FL Cell Imaging System, Thermo Fisher Scientific) at 0 and 24 h. The wound area and quantitative data analyses were performed using the Image J software (NIH, Version 1.8.0_172, Bethesda, MD, USA).

### 2.11. Statistical Methods

All statistical analyses were performed using the GraphPad Prism version 9 (GraphPad, San Diego, CA, USA). Statistical significance was determined using a *t*-test (two-tailed, equal variance). Error bars represent the mean ± SD. Statistical significance was established at *p* < 0.05.

## 3. Results

### 3.1. UCHL1 Expression in Human Uterine Leiomyoma Significantly Increased Compared to Expression in the Myometrium

The immunohistochemical analyses revealed that UCHL1 was more positive in leiomyoma tissues when compared to myometrial tissues (Figure 1A). Quantitative real-time PCR results also confirmed that the UCHL1 mRNA expression level significantly increased in leiomyoma when compared to myometrial tissues (Figure 1B). Furthermore, we analyzed protein expression by Western blotting and found that the protein expression level of UCHL1 significantly increased in leiomyoma (Figure 1C).

### 3.2. UCHL1 Silencing and Inhibition Downregulated the Collagen Production in Leiomyoma

To explore the effect of UCHL1 silencing and inhibition in leiomyoma and myometrial cells, we detected the expression of COL1A1 and COL3A1 mRNA using quantitative real-time PCR (Figure 2). The downregulation of COL1A1 and COL3A1 was not observed in myometrial cells by inhibition of UCHL1 (Figure 2A,B), and expression was markedly downregulated in leiomyoma cells (Figure 2C,D), which is consistent with these findings. The expression of COL1A1 and COL3A1 did not decrease in myometrial cells in UCHL1 shRNA transduction (Figure 2F,G). The expression of COL1A1 and COL3A1 significantly decreased in leiomyoma cells (Figure 2I,J). These results suggest that inhibition of UCHL1 suppresses abnormal collagen secretion in leiomyoma cells.

### 3.3. Effects of UCHL1 Knockdown on Collagen Gel Contraction and Wound-Healing Assay

The collagen gel contraction assay was performed to determine collagen gel contractility in UCHL1 knockdown leiomyoma and myometrial cells. After incubation for 96 h, collagen gel contraction showed a significantly lower decrease in UCHL1 shRNA of leiomyoma cells than in the control group. However, no significant differences were observed between the UCHL1 shRNA group and the control group in myometrial cells (Figure 3). In the wound-healing assay, wound closure in the UCHL1 shRNA group significantly decreased in leiomyoma and myometrial cells (Figure 4) by suppressing UCHL1 expression.

## 4. Discussion

Our study has shown that the level of UCHL1 in leiomyomas is higher than that in the matched myometrium. The pathogenesis of uterine leiomyoma is complex [18,19,20]. Under the action of various cytokines, leiomyoma cells secrete type I and type III collagen, leading to collagen deposition [7,21]. UCHL1 is expressed mainly in the kidneys, nervous, and reproductive systems. Additionally, UCHL1 controls cell death, differentiation, and proliferation [22]. The function of UCHL1 in the etiology of uterine leiomyomas is currently poorly understood. UCHL1 inhibitors have been specifically developed, however they have not yet received clinical approval [11]. Furthermore, according to our results, leiomyoma cell activity significantly decreased with the UCHL1-specific inhibitor LDN57444. LDN is used to treat a variety of cellular and animal disorders, including B-cell lymphoma, breast cancer, lung cancer, and nasopharyngeal carcinoma [23,24]. When compared to myometrial cells, UCHL1 mRNA and protein levels considerably increased in leiomyoma cells, and collagen levels decreased with the UCHL1-specific inhibitor LDN57444. These findings imply that UCHL1 may represent a feasible therapeutic target for the treatment of uterine leiomyoma.

It has been shown that the ubiquitin-proteasome system is crucial for the development of uterine leiomyomas [25]. One of the most crucial parts of the deubiquitination system, UCHL1, was first discovered as a neuronal protein [9,10]. According to some previous studies, UCHL1 expression significantly increases in some cancer cells [26,27] and decreases in the brains of Parkinson’s and Alzheimer’s disease patients [28]. There have been reports of changes in UCHL1 expression in animal hearts. For instance, following pulmonary artery constriction, the expression of ubiquitin increased in the right ventricle and UCHL1 levels were strikingly elevated in infarcted hearts [10,29,30]. In addition to the heart, uterine leiomyomas have a lower oxygen tension than the adjacent myometrium [31]. One hypothesis is that myometrial contraction and vasoconstriction during the menstrual cycle create a hypoxic environment in the uterus [32,33,34]. The similarity between the cardiac and uterine hypoxia findings may explain why UCHL1 is upregulated in uterine leiomyomas. The myometrial stem cell population, which grows preferentially in hypoxic environments, can become leiomyoma stem cells [33]. Therefore, UCHL1 and myometrial hypoperfusion have been suggested to contribute to leiomyomas’ development [35].

The deubiquitinating enzyme UCHL1 has a number of biological functions, including stabilizing monoubiquitin and cleaving its C-terminus of small substrates [10,36]. A number of proteins, including Jun-activating binding protein-1, HIF1, IκB-α, p27, p21, and p53, that are linked to cancer, neurodegenerative diseases, inflammatory response, oxidative stress, skeletal muscle development, and uterine fibroblast proliferation, are regulated by UCHL1 [23,37,38,39,40]. In cultured leiomyoma and myometrial cells, UCHL1 is known to be upregulated at least two-fold compared to in these tissues [41]. We utilized small-molecule methods to limit UCHL1 function to examine the impact of this enzyme’s activity (Figure 2). The UCHL1 hydrolase activity is potently, irreversibly, competitively, and actively site-directedly inhibited by the drug LDN57444. LDN57444 treatment for animals has reduced lung metastasis from distant tumors and hepatic fibrosis [23,42].

Furthermore, UCHL1 therapy reduced collagen gel contraction and wound healing in leiomyoma cells (Figure 3 and Figure 4). These findings imply that UCHL1 may restore dysregulated mechanical signaling in leiomyomas. We found that the inhibition of UCHL1 reduced components of the extracellular matrix, including collagen 1A1 and 3A1, in leiomyoma cells. Collagens, which constitute majority of the extracellular matrix, excessively accumulate in uterine leiomyomas. In addition to wound healing and gel contraction, the extracellular matrix components are crucial in controlling adhesion, proliferation, migration, growth, differentiation, and survival [43,44,45]. This mechanism implies that the aberrant mechanotransduction pathway is ameliorated, at least in part, by suppressing UCHL1 in uterine leiomyomas. Various cancers, including colorectal and ovarian cancers, head and neck cancers, pancreatic cancer, and hepatocellular, gastric, and esophageal carcinomas, have been linked to epigenetic silencing of the UCHL1 gene [46,47,48,49]. Additionally, UCHL1 has been reported to inhibit cell proliferation, induce apoptosis, and stabilize p53 to slow the development of cancer cells [48]. These findings imply that UCHL1 expression varies between cancer cells and leiomyoma cells, which may account for the biological differences between the two types of cells.

Our most intriguing observation may be the fact that a pharmacological inhibitor (LDN57444) with specificity for UCHL1 was able to decrease collagen formation in leiomyoma damage. A new therapeutic approach involves targeting the ubiquitin proteasome system, and deubiquitylating enzymes are becoming more prominent as pharmacological targets [50]. LDN57444 is a reversible, competitive proteasome inhibitor for UCHL1, with a 28-fold greater selectivity than UCHL3 (Ki = 0.40 M, IC_50_ = 0.88 M). LDN57444 is currently mostly used as a tool inhibitor in UCHL1 studies using disease models [23,42,51,52]. Intriguing therapeutic possibilities for the pharmacological targeting of deubiquitination in leiomyomas were highlighted in this study, and they need further investigation. We also argue that it is time to take a closer look at the function of the ubiquitin system and its regulatory enzymes in the biology of uterine leiomyomas, wherein they are likely to play a significant role in tumor growth.

## 5. Conclusions

UCHL1 is one of the molecules that could have a role in the development of uterine leiomyomas in humans.

## Figures and Tables

**Figure 1 biomolecules-13-00193-f001:**
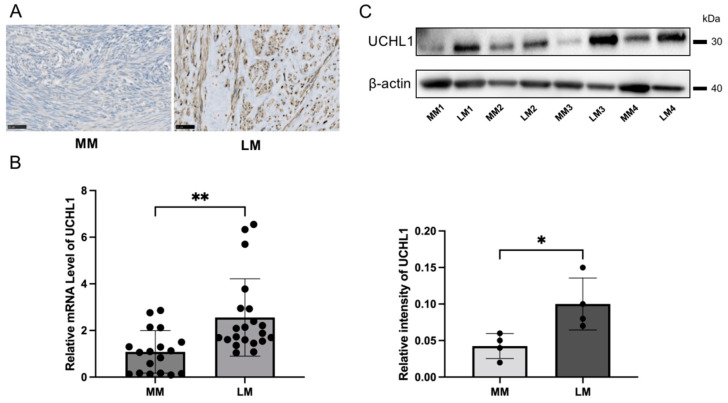
UCHL1 was highly expressed in human leiomyoma. (**A**–**C**) Strong expression of UCHL1 was observed by (**A**) immunohistochemistry, (**B**) real-time PCR, and (**C**) Western blotting in leiomyoma. The scale bar shows 50 µm. The bar shows the standard error (* *p* < 0.05, ** *p* < 0.01).

**Figure 2 biomolecules-13-00193-f002:**
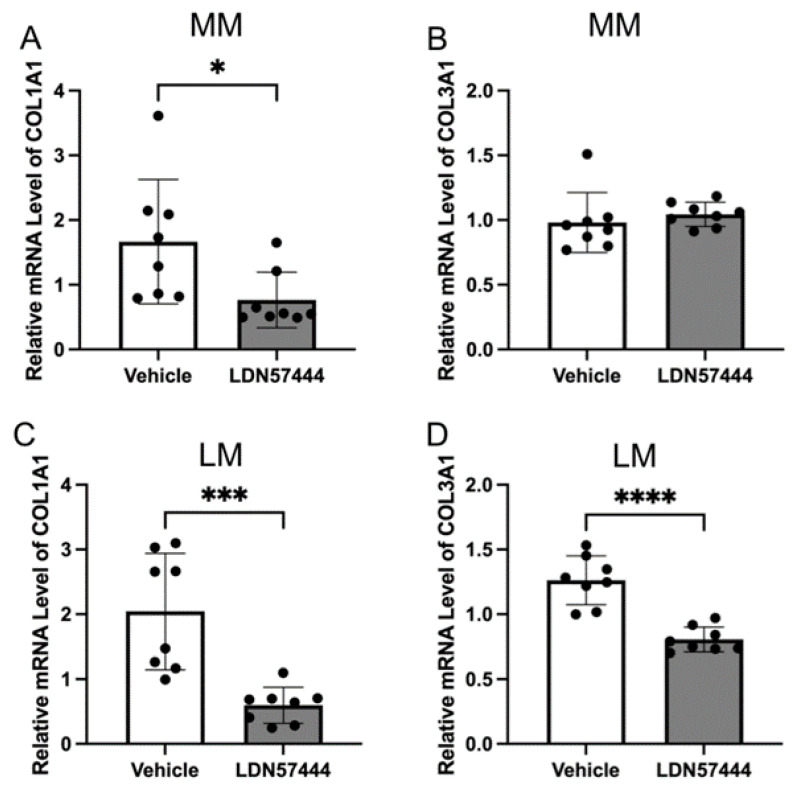
Inhibition of UCHL1 resulted in decreased collagen production in leiomyoma. mRNA expressions of COL1A1 and COL3A1 were analyzed by quantitative real-time PCR. (**A**) Downregulation of COL1A1 was observed by LDN57444 in myometrial cells. (**B**) No downregulation of COL3A1 was observed in myometrial cells. (**C**,**D**) The expression of COL1A1 and COL3A1 was downregulated by LDN57444 in leiomyoma cells. (**E**,**H**) UCHL1 knockdown efficiency was established by quantitative real-time PCR analysis of UCHL1. (**F**,**G**) The expression of COL1A1 and COL3A1 did not decrease in myometrial cells in UCHL1 shRNA transduction. (**I**,**J**) The expression of COL1A1 and COL3A1 significantly decreased in leiomyoma cells in UCHL1 shRNA transduction (* *p* < 0.05, *** *p* < 0.001, **** *p* < 0.0001).

**Figure 3 biomolecules-13-00193-f003:**
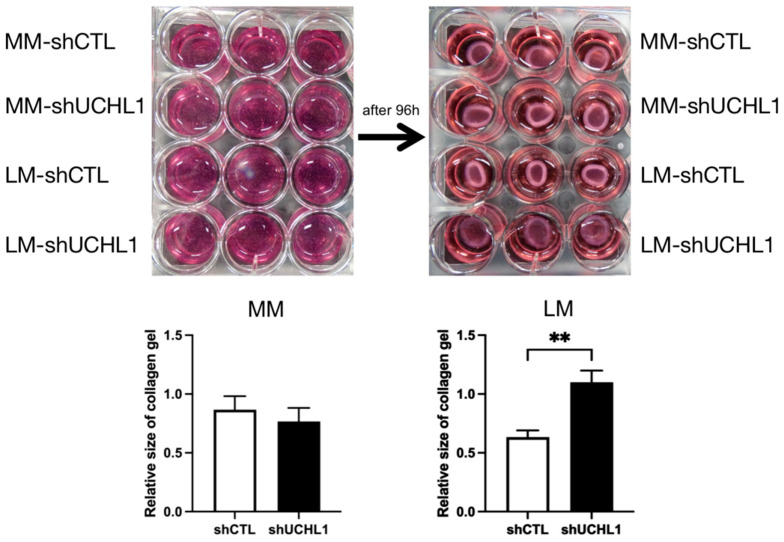
The ability of collagen gel contraction by leiomyoma cells was controlled by UCHL1. Collagen gel contraction decreased in UCHL1 shRNA of leiomyoma cells when compared to that of the control group. This finding was not observed in myometrial cells (** *p* < 0.01).

**Figure 4 biomolecules-13-00193-f004:**
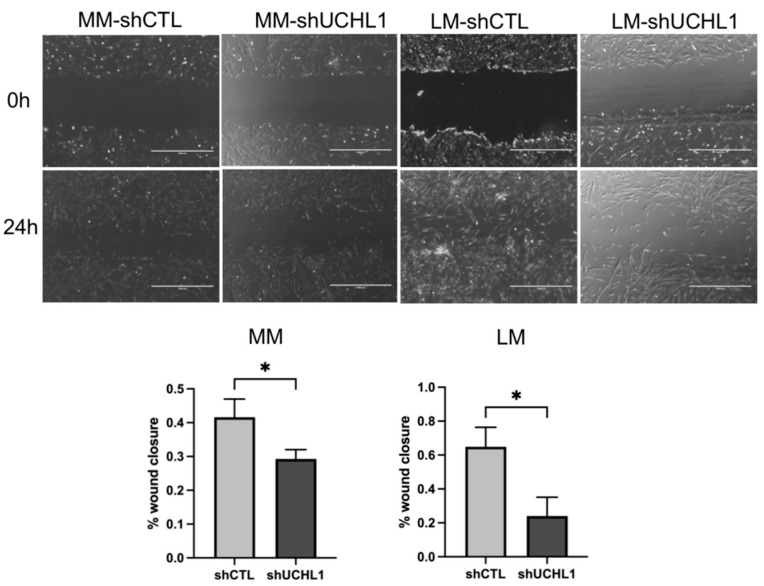
The wound-healing assay with or without UCHL1 knockdown. The wound closure of the UCHL1 shRNA group was significantly decreased in leiomyoma and myometrial cells. The scale bar shows 1000 µm (* *p* < 0.05).

**Table 1 biomolecules-13-00193-t001:** Primer pairs used for quantitative real-time PCR.

Gene Name	Primer Sets
UCHL1	5′-TGGATGGCCACCTCTATGAAC-3′ 5′-CTTGCTCACGCTCGGTGAAT-3′
COL1A1	5′-GGTGAACAGGGTGTTCCTGGAGAC-3′ 5′-AGCATCACCCTTAGCACCATCGTT-3′
COL3A1	5′-ATTATTTTGGCACAACAGGAAGCT-3′ 5′-TCCGCATAGGACTGACCAAGAT-3′
ACTB	5′-CCAACCGCGAGAAGATGA-3′ 5′-CCAGAGGCGTACAGGGATAG-3′

**Table 2 biomolecules-13-00193-t002:** Primary antibodies and suppliers used.

Antibodies	Isotype	Supplier
UCHL1	mouse IgG2A	R&D Systems (MAB6007)
β-actin (C4)	mouse IgG1κ	Millipore (MAB1501R)
HMGA2	rabbit IgG	Abcam (ab97276)
Desmin	mouse IgG	Dako (D33)
Vimentin	mouse IgG1	Merck Millipore (MAB3400)
ACTA2	mouse IgG2a	Dako (M085101)

## Data Availability

Not applicable.

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
