# Peer review of "Pivotal Role of Ubiquitin Carboxyl-Terminal Hydrolase L1 (UCHL1) in Uterine Leiomyoma"

_biomolecules, 2023, doi:10.3390/biom13020193_

Round 1

Reviewer 1 Report

The study “Pivotal role of ubiquitin carboxyl-terminal hydrolase L1 (UCHL1) in uterine leiomyoma” by Tomoo Suzuki and colleagues is undoubtedly of great interest. The discovery of potential molecular targets involved in the pathogenesis of uterine leiomyoma (UL) opens up prospects for the development of a new effective treatment method. However, reading this work raised several questions.

MAJOR

1. The authors should better describe the biomaterial used in the study unless the work is suffering from methodological flaw. It is completely unclear what kind of biomaterial was used? How the UL and myometrium cultures have been obtained? The authors referenced work of Abdelmaksoud-Dammak et al., 2016, but it is devoted to colorectal cancer and no information on UL cells is present.

2. It should be noted that UL is not a homogeneous disease, so the biomaterial must be carefully characterized before main experiments. No information is given on driver mutations in the studied biomaterial.  Modern data convincingly indicate that mutations in MED12 and in HMGA2 genes are the most frequent driver mutations in UL, also chromosomal deletions and translocations often appear in the UL (Shtykalova et al., 2021). Mutation in MED12 gene coincides with a severely reduced ability of UL cells to grow in vitro, which leads to their rapid disappearance in primary cell culture and casts doubt on the usefulness of existent UL cellular models for biomedical research (Markowski et al., 2014). New protocols are developed to prevent loss of driver mutations in MED12 and HMGA2 genes (Shved et al., 2022). The authors should add information on driver mutations in UL used in their work (both tissue and cell culture). The information on cell passages of UL cultures in time of experiment also should be included.

3. How many patients and, respectively, how many primary UL cell cultures were used in the work? Are they all characterized of the same level of UCHL1 expression? Are there any correlations with driver mutations (e.g., MED12, HMGA2)?

4. According to the current data and information given in Introduction, UCHL1 is not a common, even specific as a target for leiomyoma gene therapy. Have you examined other targets beside UCHL1?

5. Line 187: “These results suggest that inhibition of UCHL1 is effective in treating leiomyoma.”  This seems to be a controversial remark, because UCHL1 is not the only factor affecting expression of COL1A1 and COL3A1 and their products. I consider authors should formulate this statement more carefully or confirm it by additional data.

Author Response

Manuscript #biomolecules-1973923R1

Title: Pivotal role of ubiquitin carboxyl-terminal hydrolase L1 (UCHL1) in uterine leiomyoma

Reviewer’s Comments to Author:

Reviewer: 1

The study “Pivotal role of ubiquitin carboxyl-terminal hydrolase L1 (UCHL1) in uterine leiomyoma” by Tomoo Suzuki and colleagues is undoubtedly of great interest. The discovery of potential molecular targets involved in the pathogenesis of uterine leiomyoma (UL) opens up prospects for the development of a new effective treatment method. However, reading this work raised several questions.

We would like to thank you for your positive comments on our manuscript; these have helped improve the overall quality of our manuscript.

  1. The authors should better describe the biomaterial used in the study unless the work is suffering from methodological flaw. It is completely unclear what kind of biomaterial was used? How the UL and myometrium cultures have been obtained? The authors referenced work of Abdelmaksoud-Dammak et al., 2016, but it is devoted to colorectal cancer and no information on UL cells is present.

Thank you for your comments. We have modified the manuscript based on the comments you provided and added details on how leiomyomas and myometrial cells were harvested and used for subsequent experiments to the Materials and Methods as follows:

Small portions of leiomyomas and matched myometrium of patients were used to isolate leiomyomas and myometrial cells, respectively. Tissues were obtained at surgery from women (37–55 years) undergoing hysterectomy, in addition to some basic endocrine information (e.g., day of menstrual cycle and parity). Written informed consent was obtained from each patient and the use of human tissue specimens was approved by the Institutional Review Board for Human Research at Tokyo Medical University (T2021-0267). None of these cases had any previous history of uterine cancer, and all samples were confirmed by histopathological examination to be free of malignancy. The myometrial tissue was cut up manually into small pieces of <1 mm3, which were then incubated for 4 h in DMEM/F12 medium (Gibco, Grand Island, NY, USA) containing 0.2% (wt/vol) collagenase (Wako), 0.05% DNase I (Invitrogen), 1% antibiotic-antimycotic mixture (Invitrogen) at 37°C on a shaker. After the shaking, the digested tissue was filtered through a sterile gauze (Hakuzo Medical, Japan) to remove undigested tissues. The isolated leiomyoma and myometrial cells were cultured in DMEM/F12 medium (Gibco) supplemented with 10% FBS and 1% glutamine in 5% CO2 at 37° C.”

  1. It should be noted that UL is not a homogeneous disease, so the biomaterial must be carefully characterized before main experiments. No information is given on driver mutations in the studied biomaterial. Modern data convincingly indicate that mutations in MED12 and in HMGA2 genes are the most frequent driver mutations in UL, also chromosomal deletions and translocations often appear in the UL (Shtykalova et al., 2021). Mutation in MED12 gene coincides with a severely reduced ability of UL cells to grow in vitro, which leads to their rapid disappearance in primary cell culture and casts doubt on the usefulness of existent UL cellular models for biomedical research (Markowski et al., 2014). New protocols are developed to prevent loss of driver mutations in MED12 and HMGA2 genes (Shved et al., 2022). The authors should add information on driver mutations in UL used in their work (both tissue and cell culture). The information on cell passages of UL cultures in time of experiment also should be included.

Thank you. We have revised the manuscript according to the reviewer’s valuable suggestion. We performed DNA sequencing and investigated the mutation status of MED12 in the 17 leiomyoma specimens used in this experiment. It was found that 6 out of the 17 (35.29%) specimens showed MED12 mutations. We have added these results as a supplementary figure.

  1. How many patients and, respectively, how many primary UL cell cultures were used in the work? Are they all characterized of the same level of UCHL1 expression? Are there any correlations with driver mutations (e.g., MED12, HMGA2)?

Thank you. We received samples from 17 patients for this study and used 12 pairs of primary leiomyoma and myometrial specimens. Interestingly, although the expression of UCHL1 in leiomyoma varied from case to case, we found that UCHL1 expression was higher in uterine leiomyoma than in normal myometrium within the same case.

  1. According to the current data and information given in Introduction, UCHL1 is not a common, even specific as a target for leiomyoma gene therapy. Have you examined other targets beside UCHL1?

Thank you. In this research, we focused on UCHL1 because it has been reported that UCHL1 expression is upregulated in abnormal fibroblasts, which may be a significant response to pathological heart remodeling and dysfunction [1, 2]. In abnormal cardiac fibrosis, UCHL1 is a key regulator that induces collagen I and CTGF expression [3]. However, it remains unclear how UCHL1 functions in the uterus. We first examined the expression of UCHL1 in normal myometrium and leiomyoma in surgically resected specimens and found that UCHL1 expression was abnormally high in leiomyoma compared with in normal uterine muscles. This was the starting point for this research.

  1. Line 187: “These results suggest that inhibition of UCHL1 is effective in treating leiomyoma.” This seems to be a controversial remark, because UCHL1 is not the only factor affecting expression of COL1A1 and COL3A1 and their products. I consider authors should formulate this statement more carefully or confirm it by additional data.

Thank you for pointing this out. According to the reviewer’s suggestion, we have revised the manuscript as follows:

 “These results suggest that inhibition of UCHL1 suppresses abnormal collagen secretion in leiomyoma cells.”

Ultimately, we would like to express our sincere gratitude towards the editors and reviewers for their positive and constructive criticism on our manuscript. The manuscript has vastly benefited from your valuable and insightful comments and suggestions. We look forward to hearing from you and would be happy to address any further concerns, if required. We hope this further pushes the manuscript closer to publication in your esteemed journal.

References

  1. Willis, M. S.; Patterson, C., Into the heart: the emerging role of the ubiquitin-proteasome system. J Mol Cell Cardiol 2006, 41, (4), 567-79.
  2. Han, X.; Zhang, Y. L.; Fu, T. T.; Li, P. B.; Cong, T.; Li, H. H., Blockage of UCHL1 activity attenuates cardiac remodeling in spontaneously hypertensive rats. Hypertens Res 2020, 43, (10), 1089-1098.
  3. Gong, Z.; Ye, Q.; Wu, J. W.; Zhou, J. L.; Kong, X. Y.; Ma, L. K., UCHL1 inhibition attenuates cardiac fibrosis via modulation of nuclear factor-kappaB signaling in fibroblasts. Eur J Pharmacol 2021, 900, 174045.

Reviewer 2 Report

The authors investigated the role of the C-terminal ubiquitin hydrolase L1 isoenzyme in uterine leiomyoma. Study results were obtained from cell culture experiments as well as gene and protein expression in tissues. Provides some evidence for the involvement of UCHL1 in the development of fibroids.

However, a few issues need to be addressed:

1)      The first sentence of the Introduction provides not an entirely true statement. The incidence of fibroids varies among different studies and countries (4.5%–68.6%) based on the type of investigation, method of diagnosis, and racial/ethnic demographics of the population studied  (references example: Evans P, Brunsell S. Uterine fibroid tumors: diagnosis and treatment. Am Fam Physician. 2007 Epidemiology of Uterine Myomas: A Review 2016, Stewart EA, Cookson CL, Gandolfo RA, Schulze-Rath R. Epidemiology of uterine fibroids: A systematic review. BJOG. 2017). Please correct this part of the Introduction.

2)      Add more information about UCHL1 function (e.g. role in the hydrolysis of NEDD8) and its possible relation to fibroids development

3)      One of the factors involved in altered gene expression is epigenetic processes. It has been demonstrated a correlation between UCHL1 expression and promoter methylation in ovarian cancer cell lines and primary ovarian cancers (Silencing of the UCHL1 gene in human colorectal and ovarian cancers 2006) . Could you discuss this issue in relation to fibroids?

4)      There is not enough information about the material. How many patients were enrolled in the study? From what part of fibroid tissue was obtained? What was the type of fibroids? What were the characteristics of patients? (drugs, disease, age ). Similar information about the controls should be provided.

5)      Why LDN57444 inhibitor was chosen? Please provide some information about other proteasome inhibitors.

6)      Is this a total number of references in the area of UCHL1 activity in fibroids? I found others that are not included (e.g. In vitro culture significantly alters gene expression profiles and reduces differences between myometrial and fibroid smooth muscle cells  2006; A novel uterine leiomyoma subtype exhibits NRF2 activation and mutations in genes associated with neddylation of the Cullin 3-RING E3 ligase 2022).

7)      I think the conclusion is too clear that UCHL1 participates in the development of fibroids. Rather, it may be one of the factors that may be involved in the process.

Author Response

Manuscript #biomolecules-1973923R1

Title: Pivotal role of ubiquitin carboxyl-terminal hydrolase L1 (UCHL1) in uterine leiomyoma

Reviewer’s Comments to Author:

Reviewer 2

The authors investigated the role of the C-terminal ubiquitin hydrolase L1 isoenzyme in uterine leiomyoma. Study results were obtained from cell culture experiments as well as gene and protein expression in tissues. Provides some evidence for the involvement of UCHL1 in the development of fibroids. However, a few issues need to be addressed:

We sincerely appreciate this opportunity to improve our manuscript’s quality and would like to thank you for the constructive comments on this work.

1) The first sentence of the Introduction provides not an entirely true statement. The incidence of fibroids varies among different studies and countries (4.5%–68.6%) based on the type of investigation, method of diagnosis, and racial/ethnic demographics of the population studied  (references example: Evans P, Brunsell S. Uterine fibroid tumors: diagnosis and treatment. Am Fam Physician. 2007 Epidemiology of Uterine Myomas: A Review 2016, Stewart EA, Cookson CL, Gandolfo RA, Schulze-Rath R. Epidemiology of uterine fibroids: A systematic review. BJOG. 2017). Please correct this part of the Introduction.

Thank you for pointing this out. We agree that there are substantial differences in the incidence of leiomyoma between studies and countries. This sentence was rewritten as follows:

 “Despite the fact that multiple studies on the epidemiology of leiomyoma have been published, estimates of the incidence and prevalence of leiomyomas vary greatly depending on the method of diagnosis and the population under study [1-3]. Additionally, a wide range of risk factors, including biological, demographic, reproductive, and lifestyle factors, have been linked to the development of leiomyoma [1, 2, 4]. Therefore, the exact incidence and prevalence of leiomyoma remain unknown.

2) Add more information about UCHL1 function (e.g., role in the hydrolysis of NEDD8) and its possible relation to fibroids development

Thank you. We have included more information on UCHL1 and NEDD8 as follows:

 “It is also well known that UCHL1 has dual specificity for ubiquitin and Neural Precursor Cell Expressed, Developmentally Down-Regulated 8 (NEDD8), a protein encoded by the NEDD8 gene and shares 60% amino acids with ubiquitin [5]. NEDD8 is removed from protein conjugates using a variety of proteases including UCHL1. NEDD8 conjugates to its target proteins by a process known as neddylation (referred to as NEDDylation). Neddylation, a post-translational modification that involves the addition of the ubiquitin-like protein NEDD8 to substrate proteins, affects a variety of biological processes, including the advancement of the cell cycle, the control of the cytoskeleton, and the development of tumors. Aberrant neddylation is known to cause leiomyomas and liver fibrosis [6, 7].

3) One of the factors involved in altered gene expression is epigenetic processes. It has been demonstrated a correlation between UCHL1 expression and promoter methylation in ovarian cancer cell lines and primary ovarian cancers (Silencing of the UCHL1 gene in human colorectal and ovarian cancers 2006) . Could you discuss this issue in relation to fibroids?

Thank you. The following concerns for UCHL1 expression in various malignancies and leiomyomas have been added to the Discussion as follows:

 “Various cancers, including colorectal and ovarian cancers, head and neck cancers, pancreatic cancer, and hepatocellular, gastric, and esophageal carcinomas, have been linked to epigenetic silencing of the UCHL1 gene [8-11]. Additionally, UCHL1 has been reported to inhibit cell proliferation, induce apoptosis, and stabilize p53 to slow the development of cancer cells [10]. These findings imply that UCHL1 expression varies between cancer cells and leiomyoma cells, which may account for the biological differences between the two types of cells.”

4) There is not enough information about the material. How many patients were enrolled in the study? From what part of fibroid tissue was obtained? What was the type of fibroids? What were the characteristics of patients? (drugs, disease, age ). Similar information about the controls should be provided.

Thank you. We included sentences regarding patients’ information as follows in the Materials and Methods. “In this study, we used material obtained from patients hospitalized in the Department of Obstetrics and Gynecology at the Tokyo Medical University Hospital. The study participants were women qualified for surgical hysterectomy of uterine leiomyoma without any abnormalities, including adenomyosis or malignancies. Seventeen premenopausal women provided tissues from their myometrium and leiomyomas.” At the time of surgery, patients were not receiving any hormone therapy.

5) Why LDN57444 inhibitor was chosen? Please provide some information about other proteasome inhibitors.

Thank you. According to the reviewer’s valuable comments, we have included corresponding sentences to the Discussion as follows:

 “Our most intriguing observation may be the fact that a pharmacological inhibitor (LDN57444) with specificity for UCHL1 was able to decrease collagen formation in leiomyoma damage. A new therapeutic approach involves targeting the ubiquitin proteasome system, since deubiquitylating enzymes are becoming more prominent as pharmacological targets [12]. LDN57444 is a reversible, competitive proteasome inhibitor of UCHL1 with a 28-fold greater selectivity than UCHL3 (Ki = 0.40 M; IC50 = 0.88 M). LDN57444 is now mostly used as a tool inhibitor in UCHL1 studies using disease models [13-16].”

6) Is this a total number of references in the area of UCHL1 activity in fibroids? I found others that are not included (e.g., In vitro culture significantly alters gene expression profiles and reduces differences between myometrial and fibroid smooth muscle cells 2006; A novel uterine leiomyoma subtype exhibits NRF2 activation and mutations in genes associated with neddylation of the Cullin 3-RING E3 ligase 2022).

Thank you. According to the reviewer’s comments, we have included these references in the revised manuscript.

7) I think the conclusion is too clear that UCHL1 participates in the development of fibroids. Rather, it may be one of the factors that may be involved in the process.

Thank you. We rewrote the conclusion as follows:

 " Our study suggests that UCHL1 is an enzyme most likely incorporated in the development of uterine leiomyomas in humans."

Ultimately, we would like to express our sincere gratitude towards the editors and reviewers for their positive and constructive criticism on our manuscript. The manuscript has vastly benefited from your valuable and insightful comments and suggestions. We look forward to hearing from you and would be happy to address any further concerns, if required. We hope this further pushes the manuscript closer to publication in your esteemed journal.

References

  1. Stewart, E. A.; Cookson, C. L.; Gandolfo, R. A.; Schulze-Rath, R., Epidemiology of uterine fibroids: a systematic review. BJOG 2017, 124, (10), 1501-1512.
  2. Sparic, R.; Mirkovic, L.; Malvasi, A.; Tinelli, A., Epidemiology of Uterine Myomas: A Review. Int J Fertil Steril 2016, 9, (4), 424-35.
  3. Evans, P.; Brunsell, S., Uterine fibroid tumors: diagnosis and treatment. Am Fam Physician 2007, 75, (10), 1503-8.
  4. Wise, L. A.; Laughlin-Tommaso, S. K., Epidemiology of Uterine Fibroids: From Menarche to Menopause. Clin Obstet Gynecol 2016, 59, (1), 2-24.
  5. Schnell, J. D.; Hicke, L., Non-traditional functions of ubiquitin and ubiquitin-binding proteins. J Biol Chem 2003, 278, (38), 35857-60.
  6. Zubiete-Franco, I.; Fernandez-Tussy, P.; Barbier-Torres, L.; Simon, J.; Fernandez-Ramos, D.; Lopitz-Otsoa, F.; Gutierrez-de Juan, V.; de Davalillo, S. L.; Duce, A. M.; Iruzubieta, P.; Taibo, D.; Crespo, J.; Caballeria, J.; Villa, E.; Aurrekoetxea, I.; Aspichueta, P.; Varela-Rey, M.; Lu, S. C.; Mato, J. M.; Beraza, N.; Delgado, T. C.; Martinez-Chantar, M. L., Deregulated neddylation in liver fibrosis. Hepatology 2017, 65, (2), 694-709.
  7. Mehine, M.; Ahvenainen, T.; Khamaiseh, S.; Harkonen, J.; Reinikka, S.; Heikkinen, T.; Ayravainen, A.; Pakarinen, P.; Harkki, P.; Pasanen, A.; Levonen, A. L.; Butzow, R.; Vahteristo, P., A novel uterine leiomyoma subtype exhibits NRF2 activation and mutations in genes associated with neddylation of the Cullin 3-RING E3 ligase. Oncogenesis 2022, 11, (1), 52.
  8. Okochi-Takada, E.; Nakazawa, K.; Wakabayashi, M.; Mori, A.; Ichimura, S.; Yasugi, T.; Ushijima, T., Silencing of the UCHL1 gene in human colorectal and ovarian cancers. Int J Cancer 2006, 119, (6), 1338-44.
  9. Tokumaru, Y.; Yamashita, K.; Osada, M.; Nomoto, S.; Sun, D. I.; Xiao, Y.; Hoque, M. O.; Westra, W. H.; Califano, J. A.; Sidransky, D., Inverse correlation between cyclin A1 hypermethylation and p53 mutation in head and neck cancer identified by reversal of epigenetic silencing. Cancer Res 2004, 64, (17), 5982-7.
  10. Yu, J.; Tao, Q.; Cheung, K. F.; Jin, H.; Poon, F. F.; Wang, X.; Li, H.; Cheng, Y. Y.; Rocken, C.; Ebert, M. P.; Chan, A. T.; Sung, J. J., Epigenetic identification of ubiquitin carboxyl-terminal hydrolase L1 as a functional tumor suppressor and biomarker for hepatocellular carcinoma and other digestive tumors. Hepatology 2008, 48, (2), 508-18.
  11. Sato, N.; Fukushima, N.; Maitra, A.; Matsubayashi, H.; Yeo, C. J.; Cameron, J. L.; Hruban, R. H.; Goggins, M., Discovery of novel targets for aberrant methylation in pancreatic carcinoma using high-throughput microarrays. Cancer Res 2003, 63, (13), 3735-42.
  12. Harrigan, J. A.; Jacq, X.; Martin, N. M.; Jackson, S. P., Deubiquitylating enzymes and drug discovery: emerging opportunities. Nat Rev Drug Discov 2018, 17, (1), 57-78.
  13. Goto, Y.; Zeng, L.; Yeom, C. J.; Zhu, Y.; Morinibu, A.; Shinomiya, K.; Kobayashi, M.; Hirota, K.; Itasaka, S.; Yoshimura, M.; Tanimoto, K.; Torii, M.; Sowa, T.; Menju, T.; Sonobe, M.; Kakeya, H.; Toi, M.; Date, H.; Hammond, E. M.; Hiraoka, M.; Harada, H., UCHL1 provides diagnostic and antimetastatic strategies due to its deubiquitinating effect on HIF-1alpha. Nat Commun 2015, 6, 6153.
  14. Wilson, C. L.; Murphy, L. B.; Leslie, J.; Kendrick, S.; French, J.; Fox, C. R.; Sheerin, N. S.; Fisher, A.; Robinson, J. H.; Tiniakos, D. G.; Gray, D. A.; Oakley, F.; Mann, D. A., Ubiquitin C-terminal hydrolase 1: A novel functional marker for liver myofibroblasts and a therapeutic target in chronic liver disease. J Hepatol 2015, 63, (6), 1421-8.
  15. Yan, C.; Huo, H.; Yang, C.; Zhang, T.; Chu, Y.; Liu, Y., Ubiquitin C-Terminal Hydrolase L1 regulates autophagy by inhibiting autophagosome formation through its deubiquitinating enzyme activity. Biochem Biophys Res Commun 2018, 497, (2), 726-733.
  16. Kobayashi, E.; Hwang, D.; Bheda-Malge, A.; Whitehurst, C. B.; Kabanov, A. V.; Kondo, S.; Aga, M.; Yoshizaki, T.; Pagano, J. S.; Sokolsky, M.; Shakelford, J., Inhibition of UCH-L1 Deubiquitinating Activity with Two Forms of LDN-57444 Has Anti-Invasive Effects in Metastatic Carcinoma Cells. Int J Mol Sci 2019, 20, (15).

Round 2

Reviewer 1 Report

Dear Authors,

I appreciate your efforts to improve the manuscript, however, not all concerns have been addressed yet. There are still some points to clarify: 

1.     Answering the question 2 you wrote that 17 leiomyomas being used in the research were analyzed for the MED12 mutations, however the experiments were held on primary cell culture, did you analyzed them also? It is known that the UL cells are losing the MED12 mutations while culturing (from passage 2 and further). What passage was used in transfection experiments?

2.     Why other UL-associated mutations (HMGA2, chromosomal aberrations) have not been tested?

3.     Due to high heterogeneity of these tumors, authors should clarify whether they used single or multiple leiomyomas?

4.     Did the cells were analyzed using smooth muscle cells (SMC) markers e.g. the oxytocin receptor (OTR) or other? The immunocytochemistry can be carried out with desmin and vimentin to make sure whether the received cells are SMC but not mainly the fibroblasts.

Author Response

Manuscript #biomolecules-1973923R2

Title: Pivotal role of ubiquitin carboxyl-terminal hydrolase L1 (UCHL1) in uterine leiomyoma

Reviewer’s Comments to Author:

Reviewer: 1

We sincerely thank you and the reviewers for the thoughtful suggestions and insights. The manuscript has significantly benefited from these insightful suggestions.

  1. Answering the question 2 you wrote that 17 leiomyomas being used in the research were analyzed for the MED12 mutations, however the experiments were held on primary cell culture, did you analyze them also? It is known that the UL cells are losing the MED12 mutations while culturing (from passage 2 and further). What passage was used in transfection experiments?

Thank you for your comment. As you pointed out, leiomyoma cell cultures show rapid disappearance of cells carrying MED12 mutations [1]. Therefore, all leiomyoma cells used in this experiment were first passage cultures. Unfortunately, there were no remaining first passage cultures of leiomyoma cells used in this research, and DNA sequences of the MED12 mutation were not performed in these first passage cultures of leiomyoma cells.

  1. Why other UL-associated mutations (HMGA2, chromosomal aberrations) have not been tested?

Thank you for your comment. MED12 and HMGA2 aberrations may account for 80–90% of all leiomyomas [2, 3]. Chromosome 6p21 (HMGA1) rearrangements have been found to co-occur with MED12 mutations as secondary alterations relative to MED12 [3, 4]. Besides, chromosome 7q alterations have been reported to co-occur with both MED12 and HMGA2 mutations, further suggesting that 7q deletions are associated with tumor progression rather than initiation [4, 5].Therefore, we tested MED12 and HMGA2 aberrations in our specimens. We performed DNA sequencing and investigated the mutation status of MED12 in the 17 leiomyoma specimens used in this experiment. It was found that 6 out of the 17 (35.29%) specimens showed MED12 mutations (Figure S1). Immunohistochemistry was used to evaluate HMGA2 overexpression. The intensity of the HMGA2 immunoreaction was classified into three groups: - = fully negative, + = low positivity, ++ = strong positivity (Figure S2). Only samples that showed strong positivity in the immunoreaction were considered to be positive. It was found that 3 out of the 17 (17.6%) specimens showed HMGA2 mutations. We have added these results as Figure S2.

  1. Due to high heterogeneity of these tumors, authors should clarify whether they used single or multiple leiomyomas?

Thank you for your comment. We used multiple leiomyomas for our research and added details to the Materials and Methods as follows: “The study participants involved women qualified for surgical treatment for hysterectomy of multiple uterine leiomyomas without adenomyosis or malignancies.”

  1. Did the cells were analyzed using smooth muscle cells (SMC) markers e.g., the oxytocin receptor (OTR) or other? The immunocytochemistry can be carried out with desmin and vimentin to make sure whether the received cells are SMC but not mainly the fibroblasts.

Thank you for your comment. We carried out immunofluorescence using anti-desmin, anti-vimentin, and anti-α-smooth muscle actin antibodies on newly isolated first passage cultures of leiomyoma cells. We have added these results in Figure S3. The manuscript has been revised as follows in response to the reviewer's suggestions in line 151: “We confirmed that freshly isolated first passage cultures of leiomyoma cells display smooth muscle markers by conducting immunofluorescence using anti-desmin, anti-vimentin, and anti-α-smooth muscle actin antibodies.”

Finally, we would like to express our sincere gratitude towards the editors and reviewers for their positive and constructive comments on our manuscript. We look forward to working with you and the reviewer to move this manuscript closer to publication in Biomolecules.

References

  1. Nadine Markowski, D.; Tadayyon, M.; Bartnitzke, S.; Belge, G.; Maria Helmke, B.; Bullerdiek, J., Cell cultures in uterine leiomyomas: rapid disappearance of cells carrying MED12 mutations. Genes Chromosomes Cancer 2014, 53, (4), 317-23.
  2. Makinen, N.; Mehine, M.; Tolvanen, J.; Kaasinen, E.; Li, Y.; Lehtonen, H. J.; Gentile, M.; Yan, J.; Enge, M.; Taipale, M.; Aavikko, M.; Katainen, R.; Virolainen, E.; Bohling, T.; Koski, T. A.; Launonen, V.; Sjoberg, J.; Taipale, J.; Vahteristo, P.; Aaltonen, L. A., MED12, the mediator complex subunit 12 gene, is mutated at high frequency in uterine leiomyomas. Science 2011, 334, (6053), 252-5.
  3. Mehine, M.; Makinen, N.; Heinonen, H. R.; Aaltonen, L. A.; Vahteristo, P., Genomics of uterine leiomyomas: insights from high-throughput sequencing. Fertil Steril 2014, 102, (3), 621-9.
  4. Markowski, D. N.; Bartnitzke, S.; Loning, T.; Drieschner, N.; Helmke, B. M.; Bullerdiek, J., MED12 mutations in uterine fibroids--their relationship to cytogenetic subgroups. Int J Cancer 2012, 131, (7), 1528-36.
  5. Stephens, P. J.; Greenman, C. D.; Fu, B.; Yang, F.; Bignell, G. R.; Mudie, L. J.; Pleasance, E. D.; Lau, K. W.; Beare, D.; Stebbings, L. A.; McLaren, S.; Lin, M. L.; McBride, D. J.; Varela, I.; Nik-Zainal, S.; Leroy, C.; Jia, M.; Menzies, A.; Butler, A. P.; Teague, J. W.; Quail, M. A.; Burton, J.; Swerdlow, H.; Carter, N. P.; Morsberger, L. A.; Iacobuzio-Donahue, C.; Follows, G. A.; Green, A. R.; Flanagan, A. M.; Stratton, M. R.; Futreal, P. A.; Campbell, P. J., Massive genomic rearrangement acquired in a single catastrophic event during cancer development. Cell 2011, 144, (1), 27-40.